# Frailty and Sarcopenia Assessment upon Hospital Admission to Internal Medicine Predicts Length of Hospital Stay and Re-Admission: A Prospective Study of 980 Patients

**DOI:** 10.3390/jcm9082659

**Published:** 2020-08-17

**Authors:** Sapir Anani, Gal Goldhaber, Adi Brom, Nir Lasman, Natia Turpashvili, Gilat Shenhav-saltzman, Chen Avaky, Liat Negru, Muhamad Agbaria, Sigalit Ariam, Doron Portal, Yishay Wasserstrum, Gad Segal

**Affiliations:** 1Internal Medicine Department “T”, Chaim Sheba Medical Center. Sackler faculty of medicine, Tel-Aviv University, Tel Aviv P.O. Box 39040, Tel Aviv 6997801, Israel; sapir.anani@sheba.health.gov.il (S.A.); gal.goldhaber@sheba.health.gov.il (G.G.); adi.brom@sheba.health.gov.il (A.B.); nir.lasman@sheba.health.gov.il (N.L.); natia.turpashvili@sheba.health.gov.il (N.T.); gilat.shenhavzaltsman@sheba.health.gov.il (G.S.-s.); chen.avaky@sheba.health.gov.il (C.A.); Liat.negru@sheba.health.gov.il (L.N.); muhamad.agbaria@sheba.health.gov.il (M.A.); sigalit.ariam@sheba.health.gov.il (S.A.); yishay.wasserstrum@sheba.health.gov.il (Y.W.); 2Internal Medicine Department, Sackler faculty of medicine, Tel-Aviv University, Tel Aviv 6997801, Israel; doron.portal@sheba.health.gov.il

**Keywords:** sarcopenia, frailty, ALT, MAMC, frail questionnaire, internal medicine

## Abstract

Background: Frailty and sarcopenia are associated with frequent hospitalizations and poor clinical outcomes in geriatric patients. Ascertaining this association for younger patients hospitalized in internal medicine departments could help better prognosticate patients in the realm of internal medicine. Methods: During a 1-year prospective study in an internal medicine department, we evaluated patients upon admission for sarcopenia and frailty. We used the FRAIL questionnaire, blood alanine-amino transferase (ALT) activity, and mid-arm muscle circumference (MAMC) measurements. Results: We recruited 980 consecutive patients upon hospital admission (median age 72 years (IQR 65–79); 56.8% males). According to the FRAIL questionnaire, 106 (10.8%) patients were robust, 368 (37.5%) pre-frail, and 506 (51.7%) were frail. The median ALT value was 19IU/L (IQR 14–28). The median MAMC value was 27.8 (IQR 25.7–30.2). Patients with low ALT activity level (<17IU/L) were frailer according to their FRAIL score (3 (IQR 2–4) vs. 2 (IQR 1–3); *p* < 0.001). Higher MAMC values were associated with higher ALT activity, both representing robustness. The rate of 30 days readmission in the whole cohort was 17.4%. Frail patients, according to the FRAIL score (FS), had a higher risk for 30 days readmission (for FS > 2, HR = 1.99; 95CI = 1.29–3.08; *p* = 0.002). Frail patients, according to low ALT activity, also had a significantly higher risk for 30 days readmission (HR = 2.22; 95CI = 1.26–3.91; *p* = 0.006). After excluding patients whose length of stay (LOS) was ≥10 days, 252 (27.5%) stayed in-hospital for 4 days or longer. Frail patients according to FS had a higher risk for LOS ≥4 days (for FS > 2, HR = 1.87; 95CI = 1.39–2.52; *p* < 0.001). Frail patients, according to low ALT activity, were also at higher risk for LOS ≥4 days (HR = 1.87; 95CI = 1.39–2.52; *p* < 0.001). MAMC values were not correlated with patients’ LOS or risk for re-admission. Conclusion: Frailty and sarcopenia upon admission to internal medicine departments are associated with longer hospitalization and increased risk for re-admission.

## 1. Introduction

### 1.1. The Need for Internal Medicine Personalization, Hospital Length-Of-Stay and Re-Admissions

The burden, both in public-health outcomes and healthcare associated expenditures, on the hospitalization resources of internal medicine departments, worldwide, is ever growing. This is the result, in developed countries, of population growth, especially in adult and old age populations [1,2]. Improvement of prognostication measures—e.g., better appreciation of hospitalization length of stay and rate of re-hospitalizations—would be therefore of the utmost importance. Improvement of such prognostication would improve the adjustment of procedures done during the hospital stay, both diagnostic and therapeutic, for baseline patients’ characteristics. In other words, such measures serve as a true personalization of the internal medicine services delivered in hospitals worldwide.

Personalization of patients’ care in internal medicine departments refers to both estimation of length-of-stay and the anticipated rate of readmissions. If both end-points’ appreciation would be better predicted, significant changes in hospital patients’ care could be made. Regarding length of hospital stay, it is known that a better triage of patients, upon admission, could decrease the rate of patient transfer to intermediate care units that is associated with increased LOS [3,4]. Hospital readmissions are a worldwide, major health-care burden, with a negative impact on patient health, public health, and healthcare systems’ associated national financial burden. Previous publications describe the heavy toll of hospital (non-surgical) re-admissions while the same researchers and others attempted various strategies for either predicting the risk for readmission and proactively reducing it (albeit with conflicting results) [5,6,7,8,9,10].

### 1.2. Sarcopenia and Frailty Assessment, In Addition to Their Prognostic Implications in the Realm of Internal Medicine

Sarcopenia and frailty are long recognized as syndromes associated with old age, with negative impact on the risk of falls, failure of in-hospital rehabilitation, higher rates of hospital admissions, and increased risk of death [11,12,13]. In the past few years, several researchers have demonstrated the importance of sarcopenia and frailty assessment in a younger patient populations, either hospitalized in the departments of internal medicine [14] or in an ambulatory setting [15,16,17,18]. Several methods for sarcopenia and frailty assessments are available, some of which are less applicable in the setting of acute, internal medicine, hospitalizations (e.g., DEXA (Dual-energy X-ray absorptiometry test), or TUG (timed up and go test)). Nevertheless, some assessments are readily available for such patients, including alanine-aminotransferase (ALT) activity level measurements [19], and short questionnaires, like FRAIL [14].

In a previous, retrospective study, we already established an association between low ALT values and high FRAIL questionnaire score, both with each other and with increased risk of mortality in the population of internal medicine patients [14]. In the current study, we aimed at prospectively assessing the value of a multi-modal sarcopenia and frailty assessment upon hospital admission, the correlation between different indices of sarcopenia and frailty and the extent to which such indices would be associated with poor short-term clinical outcomes. Patients which, unfortunately, were already deemed to have poor prognosis upon admission (as detailed later in the ‘methods’ section) were excluded from this study.

## 2. Methods

### 2.1. Patients and Methods

After approval by an institutional review board, we recruited consecutive patients according to the following inclusion criteria: (a) Age was within the range of 55 to 85 years. We intentionally excluded the oldest (over 85 years of age) patients for whom frailty assessment is redundant. (b) We used patients admitted to the internal medicine department via the emergency department as acute in-patients. We excluded (a) advanced cancer patients (stage IV malignancy), (b) chronic obstructive pulmonary disease (COPD) patients classified as GOLD stage IV, (c) progressive dementia patients, (d) patients admitted due to diabetic foot and/or severe peripheral vascular disease, (e) congestive heart failure (CHF) patients classified as NYHA stages III or IV, (f) patients with a history of major stroke, (g) patients classified as having advanced/end-stage cirrhosis, (h) patients classified as bedridden for any cause, (i) patients necessitating mechanical ventilation upon admission, and (j) Patients hospitalized during the past 30 days. None of the aforementioned patients would benefit from diagnosis as frail since their prognosis is poor to begin with. Patients with ALT values greater than 40 IU/L were excluded from analysis (regarding ALT, as detailed later).

Upon admission, all eligible patients went through the following assessments:

#### 2.1.1. FRAIL Questionnaire

Either as part of more comprehensive indices or as stand-alone measurements, there are several validated questionnaires serving as assessment tools for frailty. One such questionnaire we were familiar with was the FRAIL questionnaire. The FRAIL questionnaire included five components: fatigue, resistance, ambulance, illnesses, and weight loss. FRAIL scores represent frail (3–5), pre-frail (1–2), and robust (0) health statuses. The FRAIL questionnaire was validated in a group of African Americans, age 49 to 65 years [20]. Morley, Malmstrom, and Miller did a cross-validation and found out that the FRAIL scale correlated significantly with IADL (instrumental activities of daily life) difficulties, handgrip strength, and one-leg stand among participants who had no baseline ADL difficulties. This correlation was demonstrated for subjects with no baseline ADL problems. The results of this longitudinal study showed that the “pre-frail” definition by FRAIL at baseline was associated with statistical significance, future ADL difficulties, worse one-leg standing, and mortality. Scoring as “frail” in the FRAIL at baseline was associated with future ADL difficulties, IADL difficulties, and increased risk of mortality. The FRAIL questionnaire is a practical and useful assessment tool that has been recommended by others [21]. A single researcher interviewed and filled the FRAIL questionnaire for all eligible patients in the current study.

#### 2.1.2. Mid-Arm Muscle Circumference (MAMC) Measurement

All anthropometric measurements are generally considered less accurate than other measurement methods and their results are highly operator-dependent. As such, anthropometric measurements necessitate usage of reliable, professional instruments, experienced testers, and application of repeated and/or multiple measurements.

There are several anthropometric measurements based on assessing LBM (lean body mass) and total body muscular mass via assimilating limb circumference [22], as well as skin-fold measurements into equations giving net muscle dimension and volume. Skin-fold measurements, applied by a caliper, are widely used for assessing percentage of fat tissue within the body. Caliper measurements of the biceps skin fold, triceps skin fold, or iliac skin fold can be placed into tables, showing normal distribution of skin-fold thickness according to gender and age. The MAMC (mid-arm muscle circumference) equation uses the upper-limb, the mid-way arm circumference measurement of the arm (MAC), and the TSF (triceps skin fold) measurement. The MAC and TSF measurement results are assimilated into the following equation: MAMC = MAC − (3.14 × TSF) [23]. The MAMC value was plotted in gender-specific tables. These tables were compiled from an historic cross-sectional sample of 19,097 white subjects, incorporated into the United States Health and Nutritional Examination Survey of 1971 to 1974 [24]. We chose the MAMC as our anthropometric “representative” since it gained relative confidence among researchers compared with other anthropometric methods for sarcopenia assessment [25,26,27,28,29,30].

In a similar manner to what we did regarding the FRAIL questionnaire, a single researcher did all MAMC measurements for all eligible patients in the current study. In our analyses, we included both raw values of MAMC measurements and percentiles drawn from the US population surveys. Since the MAMC had never been validated in the Israeli population, we preferred not to rely only on population percentile normalizations.

#### 2.1.3. ALT Measurement

Alanine amino transferase (ALT; also known as SGPT, serum glutamic pyruvic transaminase) is the enzyme responsible for reversible transamination between alanine and 2-oxoglutarate to generate pyruvate and glutamate. As such, this fundamental enzyme plays a key role in the intermediary metabolism of glucose and amino acids [31,32]. Since ALT activity in the liver is about 3000 times as high as in the serum, its main purpose in clinical testing is to rule out hepatocellular injury. The amount of ALT in tissues other than the liver, like the skeletal muscle tissue, is much lower. ALT activity levels are significantly decreased among end stage renal disease (ESRD) patients treated by hemodialysis. Moreover, ALT activity levels are lower in patients taking medications that involve/recruit the catalytic activity of P-5-P, thereby lowering ALT activity, dependent of this co-enzyme (such as dopaminergic medications used for Parkinson’s disease). The upper limit of normal (ULN) for ALT peripheral blood activity is approximately 40 IU. Above this level of activity in the blood, we assume that ALT pours out of cellular tissues. Consequently, patients with such ALT measurements were excluded from analysis whenever incorporating ALT patients’ values. There are several comprehensive publications describing the association between decreased level of ALT activity in the peripheral blood, sarcopenia, frailty, and increased risk of all-cause mortality in middle-aged, heterogeneous populations [13,17,18,19,33,34,35,36].

### 2.2. Statistical Analysis

All variables were described according to their properties. Categorical variables were reported in frequencies and percentages, and the difference between groups was tested with the Chi-square test. Continuous variables were explored using a histogram plot and the Shapiro-Wilk test. Variables found to have a normal distribution were reported as mean and standard deviation values, and the differences between groups were tested with the *t*-test method. Continuous variables that did not have a normal distribution were reported as median and interquartile ranges (IQR, 25th–75th percentiles), and the difference between groups was tested with the Mann–Whitney U test.

We preformed regressions models for 2 outcomes: a Cox-regression for 30-day hospital readmission, and a logistic regression model for prolonged length of stay, defined as hospitalizations lasting 3–10 days (longer hospitalizations were excluded, since these were mostly due to non-medical issues). The model was adjusted for age, gender, diabetes mellitus, chronic kidney disease, an active malignancy, the FRAIL score, and serum ALT.

The statistical analysis was carried out with the use of R version 3.6.1 software (The R Foundation, Boston, MA, USA) and R-studio 1.2.5001 (R Studio, Inc., Boston, MA, USA).

## 3. Results

Over a 12 month duration, we recruited 980 consecutive patients that were eligible according to the inclusion criteria and signed an informed consent. Table 1 present patients’ characteristics, according to their status of 30 day re-admission. The median age was 72 years (IQR 65–79) with 56.8% male patients. The following characteristics were found to be significantly different between patients who were re-admitted during 30 days after discharge and those who did not. Amongst re-admitted patients, there was a longer length of hospital stay (median LOS 3 (IQR; 2.00, 5.00) vs. 2 days (IQR; 1.00, 4.00), *p* = 0.015). In addition, re-admitted patients were more likely to come from nursing homes (7.6% vs. 2%, *p* = 0.004 (for the overall difference in outpatients’ settings)). Cirrhosis was also more prevalent amongst re-admitted patients (3.3% vs. 0.5%, *p* = 0.017). Laboratory parameters that were significantly different included lower hemoglobin concentration (11.8 g/dL vs. 12.8 g/dL, *p* < 0.001) and higher creatinine concentration (1.08 mg/dL vs. 0.96 mg/dL, *p* = 0.048) amongst patients with a 30-days re-admission. Patients’ characteristics associated with sarcopenia and frailty that were found to be significantly different in patients with or without a 30-days re-admission event were a lower level of ALT blood activity (median 11IU vs. 14IU, *p* = 0.005) and a higher frail score (median 3 vs. 2.5, *p* < 0.001). MAMC values were not significantly different between these two groups of patients.

Table 2 present patients’ characteristics according to the length of hospital stay. After we excluded patients hospitalized for more than 14 days (many of whom had socioeconomic rather than medical barriers for discharge) amongst patients who had a longer LOS (over 3 days duration), several characteristics were significantly different. These patients were older (median age 73 vs. 71, *p* = 0.008), their 30 day re-admission rates were higher (12.2% vs. 7.5%, *p* = 0.019), they had lower Norton and Morse scores (18 vs. 19, *p* < 0.001 and 7 vs. 6, *p* = 0.001, respectively), and a lower proportion were married (59.9% vs. 66.9%, *p* = 0.027). Laboratory parameters were also significantly different between these patients groups. Patients with longer hospital stay had lower hemoglobin concentration (12.2 g/dL vs. 13.1 g/dL, *p* < 0.001), higher creatinine concentrations (1.05 mg/dL vs. 0.92 mg/dL, *p* < 0.001), higher urea concentration (48 mg/dL vs. 41 mg/dL, *p* < 0.001), lower albumin concentration (3.7 g/dL vs. 3.9 g/dL, *p* < 0.001), and higher CRP concentration (26.3 mg/L vs. 7.2 mg/L, *p* < 0.001). Patients’ characteristics associated with sarcopenia and frailty that were found to be significantly different in patients with longer hospitalizations were lower level of ALT (median 12IU vs. 14IU, *p* < 0.001) and higher FRAIL score (median 3 vs. 2, *p* < 0.001). MAMC values were not significantly different between these two groups of patients.

Two of the measures for sarcopenia and frailty were used in this study. ALT activity measurement in the peripheral blood and the FRAIL questionnaire were not only found to be associated with higher risks for longer hospitalization and 30 days re-admission but were also found to be in correlation with each other. As presented in Table 3 and Figure 1, both ALT and frailty scores (FS) had statistically significant correlation with each other and with the MAMC measurements, although the latter was not associated with clinical outcomes. Figure 2 show a survival curve for 30 days readmission events according to FRAIL questionnaire classes (*p* = 0.0019) and Figure 3 show a survival curve for 30 days readmission events according to ALT activity lower than 12 IU (median value associated with longer hospitalizations) (*p* < 0.0001).

In the adjusted model for 30-day readmission, ALT activity levels under 12 IU and a FRAIL score of 3 or greater were predictive for 30-day readmission (HR 2.06 (95% CI 1.34–3.16) and HR 1.59 (95% CI 1.02–2.48), respectively).

In a multivariate model for prolonged hospital stay (3 - 14 days), lower ALT levels and a higher FRAIL score were the only significant predictive factors (HR 1.33 (95% CI 1.00–1.76) and HR 1.55 (95%CI 1.18–2.04), respectively).

In a subgroup analysis described in Figure 4, a lower ALT was significantly associated with 30 day readmission across most subgroups, except for a higher serum albumin level defined as >3.5 g/dL (*n* = 250, HR 1.8 (95% CI 0.84–3.90)), with no significant interactions between ALT levels and the various stratifying variables.

## 4. Discussion

Immense efforts, in time, work force, and resources are invested in making hospitalization in internal medicine department better for the sake of patients, stability, and quality of staff and in order to diminish the ever-growing healthcare associated expenses. Many of these efforts delineate the need to reduce re-admissions and shorten hospital length-of-stay. We suggest that personalization of internal medicine is the key for achieving the aforementioned targets. In contrast to precise medicine, in which the disease is better characterized and targeted (as is the case with modern oncology), we believe that personal medicine, in the realm of internal medicine, should target sarcopenia and frailty. As stated earlier, these syndromes are well described and attended in the fields of geriatrics and gerontology. Preceding old age, by identifying middle-aged patients as frail, could better their prognostication and make their in-hospital management, more accurate and more personalized. In the current study, the age of patients was a derivate of our inclusion and inclusion criteria. We included patients admitted to internal medicine (rather than geriatrics) and excluded patients who had obvious poor prognosis, for whom the assessment of sarcopenia and frailty would be redundant.

Of the many available methods for sarcopenia and frailty assessment, we chose, for the current study, three potentially complimentary approaches: one, a phenotypic assessment reflecting the consequences of frailty in daily life (the FRAIL questionnaire). The second approach addresses the sarcopenic phenotype in a more mechanistic way, i.e., the measurement of skeletal mass in conjugation with the amount of sub-cutaneous fat (the MAMC measurement, including the MAC and the TSF measurements). Last, but not least, the ALT activity measurement also addresses the net skeletal mass in the body, avoiding the bias of anthropometric measurements and readily available for all hospitalized patients, as well as the majority of community dwelling population. The relative advantages and disadvantages of each methods were described earlier. Taken together, these three assessment methods give good coverage of the fundamental characteristics of both sarcopenia and frailty. MAMC and ALT reflect both local and total body lean muscle mass. FRAIL is an established tool for frailty assessment.

In light of our results, the following conclusions were derived: (A) ALT measurements and FRAIL questionnaire results can serve as easy to get assessment tools, predicting the risk for re-admission and longer length of hospital stay. Both tools were found to have significantly statistical correlation with these clinical outcomes and with each other. (B) The MAMC measurement is not recommended as a tool for sarcopenia and frailty assessment in the setting of internal medicine department.

It is advised that, in the future, two main research aims should be sought: (A) Longer follow-up should clear the question whether sarcopenia and frailty, according to FRAIL and ALT assessments, are also indicative of worse, long-term clinical outcomes (such as 5-years survival). (B) In-hospital interventions, there is the potential to achieve better clinical outcomes in the face of sarcopenia and frailty. For example, would diabetic patients enjoy better outcomes when their glycemic control is tighter or looser in face of their being frail? Would longer intravenous antibiotics be given to frail, middle aged patients prior to discharge? Should robust patients be directed to hospital-at-home settings after initial triage and frailty assessment in the emergency department? These questions and more should be the aim of future clinical investigations. Achieving these future goals would be considered as the true embodiment of personalized-internal medicine.

## 5. Limitations

This was a single-center study. Therefore, it is possible that a selection-bias has influenced our results. In addition, long-term clinical outcomes were not reported at the time of data analysis.

## Figures and Tables

**Figure 1 jcm-09-02659-f001:**
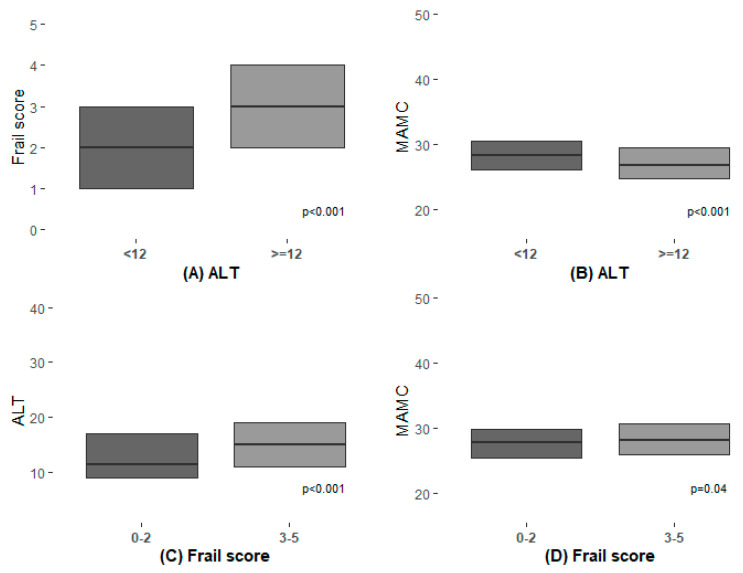
Associations between different sarcopenia and frailty indices.

**Figure 2 jcm-09-02659-f002:**
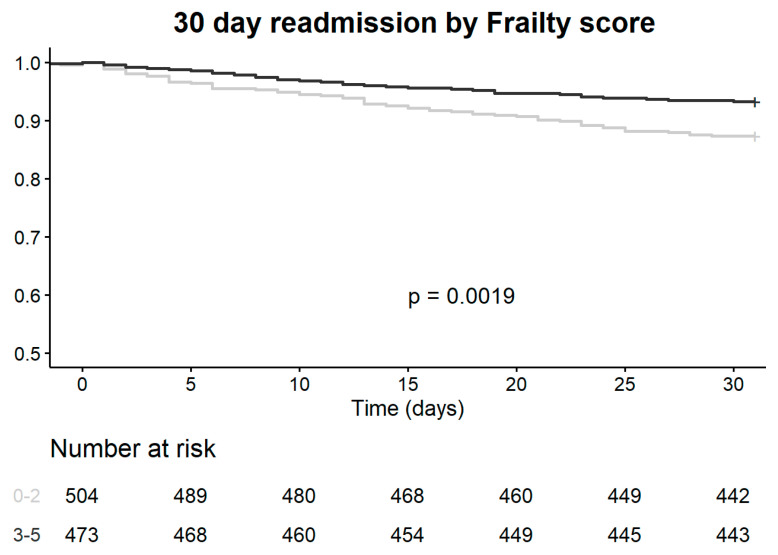
30 days readmission according to FRAIL questionnaire classes.

**Figure 3 jcm-09-02659-f003:**
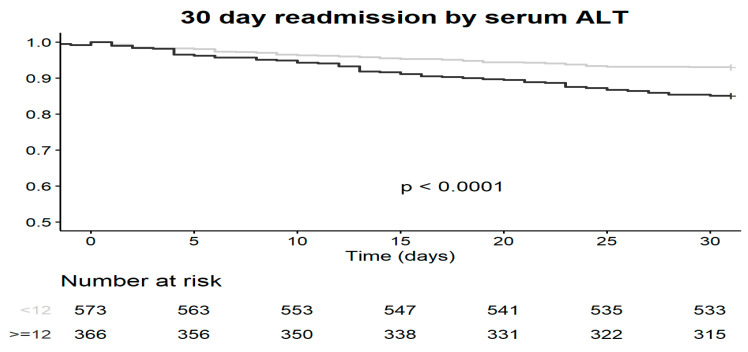
30-days readmission according to serum ALT activity.

**Figure 4 jcm-09-02659-f004:**
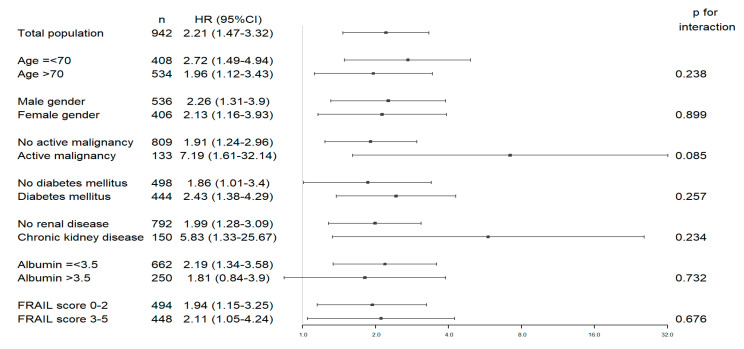
Sub-group analysis for lower ALT values and selected patients’ characteristics.

**Table 1 jcm-09-02659-t001:** Patients’ characteristics according to 30-day readmission status.

	30-Day Readmission Status		
Patients’ Characteristics	No (*n* = 888)	Yes (*n* = 92)	*p*
LOS; days (median (IQR))	2.00 (1.00, 4.00)	3.00 (2.00, 5.00)	0.015
Patient Demographics			
Female gender, *n* (%)	380 (42.8)	43 (46.7)	0.537
Age (Years), (median (IQR))	72.00 (65.00, 79.00)	74.00 (65.75, 79.50)	0.344
BMI (median (IQR))	27.12 (24.30, 31.22)	26.93 (23.51, 30.79)	0.591
Norton score (median (IQR))	19.00 (17.00, 20.00)	18.00 (17.00, 19.00)	0.167
Married, *n* (%)	573 (64.5)	49 (53.3)	0.14
Outpatient setting, before admission; *n* (%)			0.005
Community	833 (93.8)	84 (91.3)	
Nursing home	18 (2.0)	7 (7.6)	
Long-term facility	15 (1.7)	1 (1.1)	
Background Diagnoses; *n* (%)			
IHD (ischemic heart disease)	303 (34.1)	41 (44.6)	0.060
CHF (congestive heart failure)	75 (8.4)	12 (13.0)	0.199
Atrial fibrillation	174 (19.6)	25 (27.2)	0.113
DM (diabetes mellitus)	406 (45.7)	52 (56.5)	0.062
Hypoglycemia during hospitalization	5 (0.8)	1 (1.6)	0.007
Status post stroke	138 (15.5)	18 (19.6)	0.393
Dementia	13 (1.5)	0 (0.0)	0.490
CRF (chronic renal failure)	138 (15.5)	14 (15.2)	1.000
Chronic obstructive pulmonary disease	174 (19.6)	20 (21.7)	0.723
Liver cirrhosis	4 (0.5)	3 (3.3)	0.017
Solid malignancy	106 (11.9)	10 (10.9)	0.895
Laboratory Parameters (median (IQR))			
HB, g/dL	12.79 (11.39, 14.12)	11.77 (10.46, 12.96)	<0.001
CREAT, mg/dL	0.96 (0.75, 1.29)	1.08 (0.77, 1.66)	0.048
Albumin, g/dL	3.80 (3.50, 4.10)	3.80 (3.50, 4.00)	0.198
Frailty Parameters			
ALT, (IU, median (IQR))	14.00 (10.00, 18.00)	11.00 (9.00, 15.50)	0.005
FRAIL questionnaire score (median (IQR))	2.50 (1.00, 3.00)	3.00 (2.00, 4.00)	<0.001
MAMC percentile (median (IQR))	87.50 (58.80, 98.00)	86.05 (53.57, 98.00)	0.536

LOS: length of stay; BMI: body mass index; MAMC: mid-arm muscle circumference.

**Table 2 jcm-09-02659-t002:** Cohort according to LOS (patients whose LOS ≥ 14 days excluded).

Patients’ Characteristics	Length of Hospital Stay ≥ 3 days		*p*
	No (*n* = 504)	Yes (*n* = 456)	
30-day readmission	38 (7.5)	56 (12.3)	0.018
Patient Demographics			
Female gender, *n* (%)	216 (42.9)	199 (43.6)	0.858
Age (Years), (median (IQR))	71.00 (64.00, 78.00)	73.00 (65.00, 80.00)	0.005
BMI (median (IQR))	27.16 (24.55, 31.20)	27.06 (23.95, 31.22)	0.648
Norton score (median (IQR])	19.00 (18.00, 20.00)]	18.00 (15.00, 19.00)	<0.001
Married, *n* (%)	337 (66.9)	275 (60.3)	0.041
Outpatient setting, before admission; *n* (%)			0.027
Community	486 (96.4)	419 (91.9)	
Nursing home	8 (1.6)	16 (3.5)	
Long-term facility	4 (0.8)	8 (1.8)	
Background Diagnoses; *n* (%)			
IHD	167 (33.1)	173 (37.9)	0.137
CHF	42 (8.3)	44 (9.6)	0.549
Atrial fibrillation	96 (19.0)	101 (22.1)	0.268
DM	223 (44.2)	223 (48.9)	0.168
Hypoglycemia during hospitalization	2 (0.5)	4 (1.3)	0.503
Status post stroke	78 (15.5)	77 (16.9)	0.614
Dementia	10 (2.0)	3 (0.7)	0.135
CRF; Chronic renal failure	67 (13.3)	81 (17.8)	0.068
Chronic obstructive pulmonary disease	92 (18.3)	98 (21.5)	0.240
Liver cirrhosis	4 (0.8)	3 (0.7)	1.000
Solid malignancy	56 (11.1)	57 (12.5)	0.571
Laboratory Parameters (median (IQR))			
HB, g/dL	13.08 (11.85, 14.33)	12.23 (10.74, 13.63)	<0.001
CREAT, mg/dL	0.92 (0.73, 1.15)	1.04 (0.79, 1.54)	<0.001
Albumin, g/dL	3.90 (3.70, 4.20)	3.70 (3.30, 4.00)	<0.001
Frailty Parameters			
ALT; (median (IQR))	14.00 (10.00, 19.00)	12.00 (9.00, 17.00)	<0.001
FRAIL questionnaire (FQ) total score	2.00 (1.00, 3.00)	3.00 (2.00, 4.00)	<0.001
MAMC percentile (median (IQR))	28.22 (25.84, 30.25)	27.70 (25.48, 29.90)	0.148

**Table 3 jcm-09-02659-t003:** Associations between different sarcopenia and frailty indices.

	***n***	**ALT < 12 IU**	***n***	**ALT ≥ 12 IU**	***p***
Frail score (median (IQR))	411	3 (2.0–4.0)	227	2 (1.0–3.0)	<0.001
MAMC (median (IQR))	411	26.81 (24.84, 29.47)	227	28.31 (26.12, 30.59)	<0.001
	***n***	**Frail Score 3–5**	***n***	**Frail Score 0–2**	***p***
ALT (median (IQR))	322	15.00 (11.00, 19.00)	316	11.50 (9.00, 17.00)	<0.001
MAMC (median (IQR))	506	28.15 (25.84, 30.62)	474	27.72 (25.41, 29.81)	0.04

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
