# Peer review of "Frailty and Sarcopenia Assessment upon Hospital Admission to Internal Medicine Predicts Length of Hospital Stay and Re-Admission: A Prospective Study of 980 Patients"

_jcm, 2020, doi:10.3390/jcm9082659_

Round 1

Reviewer 1 Report

Thank you for the opportunity of reviewing this manuscript, which studies the impact of frailty and sarcopenia in the risk of readmissions in Internal Medicine areas.

There are some major concerns and minor questions, which could improve the manuscript:

A) Major questions:

  1. Sarcopenia definition: Authors speak of sarcopenia, but they do not use any of the validated tools for diagnosing it. They only measure MAMC and ALT. The current definition os sarcopenia (Cruz-Jentoft et al , 2019) needs the presence of a low muscle strength and a low lean muscle mass. Authors should retire all results related to sarcopenia, or incorporate this approach if they have the data. 

2. Inclusion and exclusion criteria. Representativeness of the sample: Authors exclude patients with malignancy, severa COPD and heart failure, stroke or dementia, cirrhosis, and those 85 y.o. or older. What is the reason for that? I mean, these are around 30-50% of internal medicine patients' population. The prevention, early detection, and treatment of both phenotypes (frailty and sarcopenia) are also useful in these patients. Excluding all these groups of patients reduces its representativeness. I suggest to include all these patients if authors have their data.

 3. LOS and sociofamiliar aspects. In the same way authors exclude patients with a high LOS, due mainly to sociofamiliar issues. Social and familiar issues are key health-status determinants, and elevated LOS induce clinical complications. Patients of both scenarios are at higher risk for developing frailty and sarcopenia. Authors should explain why excluding them.  I suggest to include them in the analysis y authors have the data.

.

B) Minor questions

  1. Results: LOS and mean age. Both are striking low. Generally internal medicine hospitalization areas have patients with a mean age of 75-80 y.o. and usual LOS is rounding 6-9 days. Authors should explain this difference. 
  2.  Mortality. It would be beneficial to detail in-hospital mortality rate for readers. Mortality, LOS and readmissions are linked concepts.
  3. Tables: Authors have excluded patients with cancer, severe COPD, Heart failure, cihrrhosis, but these chronic conditions appear in Tables 1 and 2. Please explain.
  4.  Cut-off thershholds. ALT cut-off of 12, and age cut-off in 70 y.o. rationale should be explained.
  5. Multivariant analysis: social and familiar issues are a core risk factor for LOS and readmissions. I suggest to include it in the multivariant models to really control confounders. 

Reviewer 2 Report

Overall, this is an interesting and well-designed study, with its sample size and prospective nature being strengths. The use of three different and diverse measurements of sarcopenia/frailty are also positives.

Please clarify the following points:

1) The abstract and introduction state that this study is focused on "younger patients", with the intent to distinguish from prior sarcopenia studies focusing on geriatric populations. However, the discussion later mentions "middle aged patients." Could the authors please clarify what their cut-offs are for the chosen age group for this study?

2) The introduction and discussion both mention the concept of more personalized care and management based on sarcopenia/frailty measures. However, the results focus on two broader outcomes, risk of re-admission and length of hospital stay. Perhaps one could make the connection between sarcopenia/frailty measures and personalized care if the measures had led to a difference in clinical outcomes related to comorbidities in the patient population (diabetes, CKD, etc). However, with the current results, it may be an overreach to associate the studied sarcopenia/frailty measures with changes in personalized care.

3) Could the authors please clarify/provide rationale for stratifying ALT activity levels as less/greater than 12IU, as well as the definition of longer hospital stay as >3 days?

4) This may be outside the scope of the study, but it would be useful to ascertain whether the combination of 2 or more of the studied sarcopenia/frailty measurements can be used to create a more accurate assessment tool for predicting risk of re-admission and length of hospital stay. In the absence of this, it may not be accurate to describe the measurements as a "multi-modal assessment" (line 74).

Regarding English language, there are several sentences which have grammatical errors and run-ons, and overall the manuscript may benefit from editing. Some typos and errors are noted below:

Line 3 (Title): “Predict” should be changed to “Predicts” (Frailty and Sarcopenia Assessment Predicts Worse Outcomes)

Line 21: Uncapitalize “Nine”

Line 46-47: It’s unclear what “patients’ nature” refers to.

Line 49: Please change “refer” to “refers”

Line 53: Please change “patients transfer” to “patient transfer”

Line 64: Considering changing to “younger patient populations”

Line 80-81: Please clarify “oldest patients” (patients older than age 85?)

Line 83: Consider “Advanced cancer” instead of “Advanced”

Line 89: “Foresee a grim prognosis” is a bit subjective in the context of developing methodology. Perhaps a reference describing these comorbidities as inherently scoring high on sarcopenia/frailty assessments would be useful.

Line 183, 200: Please correct “Mid-are muscle circumference”

Line 205: Please change “later” to “latter”

Line 246: Please change “boas” to “bias”

Round 2

Reviewer 1 Report

Thank you for your responses.

Author Response

Dear reviewer, 

I thank you very much for your work. 

Gad Segal, MD